# PAINT: PARALLEL-IN-TIME NEURAL TWINS FOR DYNAMICAL SYSTEM RECONSTRUCTION

## ABSTRACT

Neural surrogates have shown great potential in simulating dynamical systems, while offering real-time capabilities. We envision *Neural Twins* as a progression of neural surrogates, aiming to create digital replicas of real systems. A neural twin consumes measurements at test time to update its state, thereby enabling context-specific decision-making. A critical property of neural twins is their ability to remain *on-trajectory*, i.e., to stay close to the true system state over time. We introduce **Pa**rallel-in-t**i**me **N**eural **T**wins (**PAINT**), an architecture-agnostic family of methods for modeling dynamical systems from measurements. PAINT trains a generative neural network to model the distribution of states parallel over time. At test time, states are predicted from measurements in a sliding window fashion. Our theoretical analysis shows that PAINT is on-trajectory, whereas autoregressive models generally are not. Empirically, we evaluate our method on a challenging two-dimensional turbulent fluid dynamics problem. The results demonstrate that PAINT stays on-trajectory and predicts system states from sparse measurements with high fidelity. These findings underscore PAINT's potential for developing neural twins that stay on-trajectory, enabling more accurate state estimation and decision-making. [1]

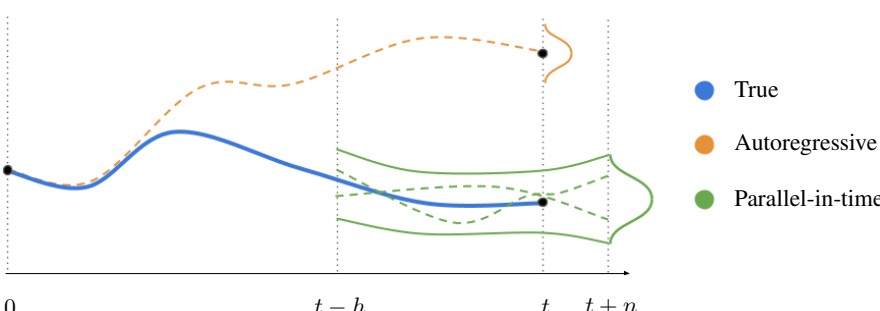

Figure 1: Real world measurements allow parallel-in-time models to stay on-trajectory, while autoregressive models may drift over time. Dashed lines represent sampled trajectories. Parallel-in-time models take a window of measurements and predict a distribution of subtrajectories which can be sampled and used for state and uncertainty estimation. Autoregressive models are prone to *over-reliance on their autoregressive state* (see Section 3.3) and typically depend on an initial state.

## 1 INTRODUCTION

Modern artificial intelligence is increasingly integrated into the modeling, monitoring, and control of complex dynamical systems. A key development in this domain are digital twins (Grieves, 2014): virtual models that run alongside a physical system, using live sensor data to update its predictions. Unlike standard simulations, a digital twin continuously adjusts its internal state to match the real system's behavior, thereby enabling better monitoring and automated decision-making.

---

[1] The code will be made public upon acceptance.

Neural surrogates emulate expensive, high-fidelity simulations and offer fast predictions while maintaining reasonable accuracy. Recent works have already demonstrated remarkable results on computational fluid dynamics (CFD) (Li et al., 2022a; Alkin et al., 2024a; Wu et al., 2024; Alkin et al., 2025), as well as on particle based systems as surrogate models for discrete element methods (Alkin et al., 2024c).

However, transforming neural surrogates into neural twins is a challenging endeavor. The current generation of neural surrogates can neither ingest nor adapt to real-time measurements during inference. Therefore, we ask the question: *How to design neural surrogates that adapt at test time using real-time measurements?*

We address this question by introducing **Pa**rallel-in-**ti**me **N**eural **T**wins (**PAINT**), a family of methods for reconstructing dynamical systems from measurements. At the core of PAINT is training a generative neural network to model the joint conditional probability of system states given the measurements. At inference, it reconstructs a sequence of states from measurements using a sliding-window approach. This makes our method interpretable and avoids any dependence on an initial condition.

Mathematically, we show that PAINT enables accurate state estimation from measurements due to being *on-trajectory*, i.e., staying close to the true trajectory even in presence of prediction errors. We also show that autoregressive models may possess this property under certain conditions (e.g. Kalman filters), but not in general. Our analysis also sheds light on a phenomenon we call *over-reliance on the autoregressive state*. It describes the problem of an autoregressive model over-relying on an off-trajectory state, whereas an ideal model would recognize such drift and reduce the state's influence on the prediction.

Empirically, we implement FlowPAINT, an instance of PAINT based on Flow Matching. Flow-PAINT is evaluated on a dataset of state-of-the-art large eddy simulations with varying Reynolds numbers in 2D. This dataset is especially challenging due to the chaotic dynamics of the turbulent fluid. The results confirm the generalization capabilities of our method.

We show – theoretically and empirically – how to design neural twins which stay on-trajectory for arbitrary many timesteps. Our main contributions are summarized as follows:

- To the best of our knowledge, we are the first to train an on-trajectory neural twin for dynamical systems. Our method is interpretable, has an inherent measure of uncertainty and takes no prior assumptions on the initial state, therefore making it more widely applicable.

- Mathematically, we show that our method is on-trajectory, i.e., stays close to the true trajectory for arbitrarily long rollouts. Our analysis also sheds light on pitfalls when using autoregressive models with sparse measurements.

- We empirically demonstrate the effectiveness of our method on a challenging 2D turbulent fluid dynamics problem.

## 2 BACKGROUND

**Notation.** Vectors and matrices are denoted in bold, e.g., $\boldsymbol{x} \in \mathbb{R}^n$ and $\boldsymbol{A} \in \mathbb{R}^{m \times n}$. General sequences such as $[k, \ldots, t]$ are denoted in brackets as $[k, t]$. We may also use brackets as subscripts over sequences, e.g., $\boldsymbol{x}_{[k,t]} = [\boldsymbol{x}_k, \ldots, \boldsymbol{x}_t]$. True or ground-truth probability distributions are denoted $p(\cdot)$, while functions or distributions parameterized by neural networks use subscripts $\theta$ for their parameters, e.g., $f_\theta$ or $p_\theta(\cdot)$.

### 2.1 DIGITAL TWINS AND DATA INTEGRATION

**Digital twins** as defined by Grieves (2014) interact with an industrial process by (a) consuming real-time data and (b) taking decisions based on the current state of the system. Ozan et al. (2025) marks a first end-to-end approach to incorporate and control a PDE with a learned policy. However as the state estimation is not interpretable, these models are hard to evaluate and can hardly be trusted for real-world production use cases. For a comprehensive introduction we refer to (Thelen et al., 2022).

**Bayesian filtering.** Updating model states by incorporating measurement data has a rich history, dating back to the seminal works on Wiener and Kalman filters (Wiener, 1949; Kalman, 1960). These types of Bayesian filters were extensively studied, leading to generalizations and extensions like the extended, unscented Kalman filters or particle methods. Bayesian filtering methods can be categorized by (1) Filtering: $p(\boldsymbol{x}_t \mid \boldsymbol{m}_{[0,t]})$, (2) Prediction: $p(\boldsymbol{x}_{t+n} \mid \boldsymbol{m}_{[0,t]})$, and (3) Smoothing: $p(\boldsymbol{x}_t \mid \boldsymbol{m}_{[0,T]})$, where $\boldsymbol{x} \in \mathbb{R}^{d_x}, \boldsymbol{m} \in \mathbb{R}^{d_m}, t \in [0, T]$ and $n \in \mathbb{N}$. We refer to Särkkä & Svensson (2023) for a comprehensive overview on classical Bayesian filtering.

With the rise of deep learning in the mid-2010s, several works married Kalman or Bayesian filters with deep neural networks (Krishnan et al., 2015; Karl et al., 2016; Kim et al., 2025). However, similar to the original Kalman filter these works usually rely on simplifying assumptions, on the model, noise or initial condition, which is in stark contrast to our method.

**Reconstructing flows from sparse measurements.** Prior art has explored flow reconstruction from sparse measurements. These measurements are usually assumed to come from Particle Tracking Velocimetry (PTV), probes in the walls or at informative locations. Some works used generative modeling techniques to reconstruct flow fields from sparse measurements for a single timestep without temporal context (Güemes et al., 2022; Cuéllar et al., 2024; Oommen et al., 2025; Kim et al., 2021; Hemant Parikh et al., 2025). Similarly, Chakraborty et al. (2025) trained an instantaneous super-resolution weather model. Several works used physics-informed neural networks (PINNs) (Raissi et al., 2019) to reconstruct an instantaneous flow field from sparse measurements (Chaurasia & Chakraborty, 2024; Hosseini & Shiri, 2024; Toscano et al., 2024), however they lack the generative modeling which is necessary to rightfully model the distribution of possible solutions. Dang et al. (2024) used spatiotemporal information for reconstruction, but they did not model the target trajectories probabilistically and therefore learn the theoretical mean of the conditional distribution. From a conceptual point of view, the closest to our concrete implementation of PAINT is Sun & Wang (2020), however their construction based on Bayesian neural networks is unnecessarily complex, did only work on simple laminar flows and is not scalable to larger settings.

## 2.2 SCALABLE NEURAL SURROGATES

**Surrogates models.** A surrogate model, or simply a surrogate (Forrester et al., 2008), is a simplified computational model designed to approximate a more intricate, computationally intensive system, such as a high-fidelity simulation or a physical experiment.

**Transformers.** The application of transformers (Vaswani et al., 2017) as surrogate models has recently taken over, building on their established efficacy in diverse scientific domains, including protein folding (Jumper et al., 2021; Abramson et al., 2024) and weather forecasting (Bi et al., 2023; Bodnar et al., 2024). These transformer-based neural surrogates are engineered to synthesize information across varying spatial locations and scales by leveraging attention mechanisms (Vaswani et al., 2017). Transformer-based surrogates represent an extension of the neural operator framework (Lu et al., 2019; 2021; Li et al., 2020b;a; Kovachki et al., 2021), specifically by integrating self-attention, cross-attention, or perceiver blocks (Jaegle et al., 2021b;a). Notable examples of these models include OFormer (Li et al., 2022b), Transolver (Wu et al., 2024), and (AB-)UPT (Alkin et al., 2024b; 2025). They have paved the way for scalable neural surrogate learning, scaling from a few thousand (Li et al., 2022a; Wu et al., 2024; Alkin et al., 2024b) to millions of mesh points (Alkin et al., 2025).

**Generative modeling of PDEs.** Over the past years, a powerful group of related generative modeling frameworks were introduced around the works of Diffusion (Sohl-Dickstein et al., 2015; Ho et al., 2020), Flow Matching (Liu et al., 2022; Lipman et al., 2022) and stochastic interpolants (Albergo & Vanden-Eijnden, 2022; Albergo et al., 2023a; Gao et al., 2024).

Generative models were used to model PDEs (Sestak et al., 2025; Kohl et al., 2023). Some works emphasize the beneficial properties of diffusion models for modeling high frequencies (Lippe et al., 2023), leading to more accurate and physically plausible solutions. Our work differs, as prior works sample *any* trajectory $p(x_{[0,t]})$, while we model $p(x_{[0,t]} \mid m_{[0,t]})$ and aim to stay on the real trajectory from measurements $m_{[0,t]}$ (see Section 3.2).

## 2.3 Computational Fluid Dynamics

The fluid mechanics problem, under the continuum assumption, is governed by the Navier–Stokes equations, a set of highly complex, second-order nonlinear PDEs. These equations are complemented by appropriate boundary and initial conditions and are for engineering problems typically solved numerically. Because turbulence in fluid flows is in most engineering applications inevitable and involves a wide range of spatial and temporal scales, direct numerical simulation (DNS) is almost never feasible in practice. Most industrial applications therefore rely on Reynolds-Averaged Navier–Stokes (RANS) approaches, which solve only for mean fields under strong modeling assumptions. Large-Eddy Simulation (LES) offers a compromise: it resolves the larger, energy-containing scales while modeling only the smallest, but remains computationally demanding (Davidson, 2015; Fröhlich, 2006).

$$\nabla \cdot \boldsymbol{u} = 0$$
$$\rho \frac{\partial \boldsymbol{u}}{\partial t} + \rho(\boldsymbol{u} \cdot \nabla)\boldsymbol{u} = -\nabla p + \mu \Delta \boldsymbol{u} + \rho \boldsymbol{f_e} \tag{1}$$

## 3 Method

### 3.1 A neural twin framework for dynamical systems

We envision neural twins in a four-step process, depicted in Figure 2. First, design knowledge of the dynamical system is used to create the simulations. This ensures that the simulations reflect physical constraints and key behaviors. Second, a diverse training dataset is generated by exploring a broad spectrum of trajectories. This dataset is used for training a neural twin. Third, at test time the neural twin dynamically incorporates real-time measurement data, allowing it to stay aligned with actual system trajectories and support real-time decision-making. Fourth, via the state estimation of the neural twin, one can optionally control the dynamical system, e.g. via rule-based methods. The approach combines domain expertise with adaptive learning, enabling both accuracy and agility in complex environments.

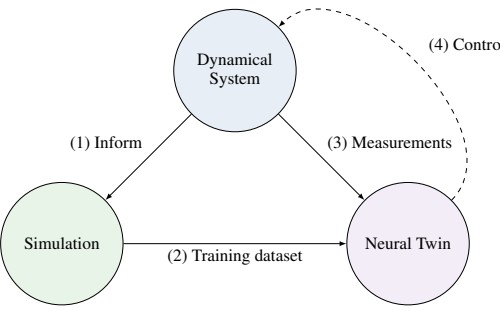

Figure 2: Interaction between a real-world dynamical system, simulations, and a neural twin. The workflow is structured in three phases: **(1)** The dynamical systems informs the design of the simulations. **(2)** Via the simulations, a diverse training dataset is generated, which is used for training the neural twin. **(3)** During inference, the neural twin integrates real-time measurement data to remain on-trajectory. **(4)** The state estimation of the neural twin can optionally be used for decision-making.

### 3.2 Parallel-in-time Neural Twins

**Problem description.** Consider a transient dynamical system where the state space is a compact subset $\mathcal{X} \subset \mathbb{R}^{d_x}, d_x \in \mathbb{N}$ and $\boldsymbol{x}_t \in \mathcal{X}$ denotes the system state at time $t$. We assume a $d_m \in \mathbb{N}$ dimensional compact measurement space $\mathcal{M} \in \mathcal{R}^{d_m}$. At each timestep $t \in \mathbb{N}$ the system provides a measurement $\boldsymbol{m}_t$ from the emission model $p_e$, where $\boldsymbol{m}_t \sim p_e(\boldsymbol{m}_t \mid \boldsymbol{x}_t)$.

**PAINT.** The goal of our generative model $p_\theta$ is to sample trajectories from $p(\boldsymbol{x}_{[0,t]} \mid \boldsymbol{m}_{[0,t]})$. At the core of our method is *joint modeling* of this distribution. In contrast, *autoregressive models* factorize this distribution which may lead to drifting (see Section 3.3).

As the number of timesteps may become very large, PAINT models the joint probability over a window (see Figure 1) of width $h$ and may predict $n$ timesteps into the future:

$$p_\theta(\boldsymbol{x}_{[t-h,t+n]} \mid \boldsymbol{m}_{[t-h,t]}). \tag{2}$$

In this setup, future measurements can inform past states to obtain temporal consistency. Furthermore, there is no reliance on the existence or an assumption about an initial condition.

PAINT is agnostic to the generative modeling framework and the neural network architecture. Common building blocks may be transformers, MLPs, CNNs (LeCun et al., 2002).

## 3.3 Theoretical Considerations

**Autoregressive models** factorize the joint distribution as a product of conditionals. Assuming the system is Markovian and conditioned on a sequence of variables $\boldsymbol{m}_{[0,t]}$, we obtain:

$$p(\boldsymbol{x}_{[0,t]} \mid \boldsymbol{m}_{[0,t]}) = p(\boldsymbol{x}_0 \mid \boldsymbol{m}_{[0,t]}) \cdot \prod_{i=1}^{t} p_\theta(\boldsymbol{x}_i \mid \boldsymbol{x}_{i-1}, \boldsymbol{m}_{[0,t]}), \tag{3}$$

where the autoregressive model learns the conditional probability $p_\theta(\boldsymbol{x}_t \mid \boldsymbol{x}_{t-1}, \boldsymbol{m}_{[0,t]})$. Notice, this does not equal $p_\theta(\boldsymbol{x}_t \mid \boldsymbol{x}_{t-1}, \boldsymbol{m}_t)$, although e.g. Kalman filters use this form.

**Autoregressive model drift.** Our analysis is leaned on the analysis of error growth of dynamical systems (Orrell, 2005) and key observations made in (Hess et al., 2023). We assume a fully differentiable neural network , which learns the probability distribution $p_\theta(\boldsymbol{x}_t \| \boldsymbol{x}_{t-1}, \boldsymbol{m}_{[0,t]})$. To sample from this probability distribution $\boldsymbol{x}_t \sim p_\theta(\boldsymbol{x}_t \| \boldsymbol{x}_{t-1}, \boldsymbol{m}_{[0,t]})$ we rewrite the neural network to a deterministic function with a stochastic input $\eta_t$:

$$\boldsymbol{x}_t = f_\theta(\boldsymbol{x}_{t-1}, \boldsymbol{m}_{[0,t]}, \eta_t) \qquad \eta_t \sim p(\eta) \tag{4}$$

The Jacobian is then defined as $\boldsymbol{J}_t := \partial f_\theta / \partial \boldsymbol{x}_t$. In contrast to Orrell (2005), this is a discrete random ordinary differential equation (RODE), i.e., an ODE with a random input at each timestep.

To show the model drift from small perturbations, we fix a path of the RODE by fixing $\eta_{[0,t]}$. For simplicity, we then write $F_\theta^t(\boldsymbol{x}_t) = f_\theta(\boldsymbol{x}_{t-1}, \boldsymbol{m}_{[0,t]}, \eta_t)$ where we absorb the measurement $\boldsymbol{m}_{[0,t]}$ and the stochastic input $\eta_t$ into the function, such that $\boldsymbol{x}_t = F_\theta^t(\boldsymbol{x}_{t-1})$. Hence for the generation of a state $\boldsymbol{x}_t$ from a previous state $\boldsymbol{x}_k$ it follows from function composition:

$$\boldsymbol{x}_t = F_\theta^t \circ F_\theta^{t-1} \circ ... \circ F_\theta^{k+1}(\boldsymbol{x}_k) = F_\theta^{[k+1,t]}(\boldsymbol{x}_k) \tag{5}$$

The first-order Taylor approximation of a small input perturbation at $t = k$ is:

$$F_\theta^{[k+1,t]}(\boldsymbol{x}_k + \epsilon) = F_\theta^{[k+1,t]}(\boldsymbol{x}_k) + \epsilon \prod_{i=k+1}^{t} J_i(\boldsymbol{x}_{i-1}) + o(\epsilon^2) \tag{6}$$

This means, a deviation at timestep $k$ induces a deviation multiplied by the *Jacobian product series* at timestep $t$. This has also been experimentally explored by Hess et al. (2023), who include visualizations of the Jacobian product series for chaotic dynamical systems.

**On-trajectory.** In the following we will formalize the notions of a model being "on-trajectory". For this, we rely on Assumption 1, which states that the predictive relevance of past measurements decays within a finite window $h$.

**Assumption 1.** *Temporal decay of measurement information: For the given dynamical system, there is a finite window size $h \in \mathbb{N}$, such that almost all information from past measurements comes from measurements within this window. Formally, for all $\epsilon > 0$ and all $t \in \mathbb{N}$ there exists an $h \ll t$ such that $\|p(\boldsymbol{x}_t \mid \boldsymbol{m}_{[t-h,t]}) - p(\boldsymbol{x}_t \mid \boldsymbol{m}_{[0,t]})\| < \epsilon$.*

Next, we proceed with our definition of on-trajectory.

**Definition 1.** *On-trajectory: Assuming unbounded model size, compute and data, a model is on-trajectory if the modeled distribution is arbitrarily close to the true distribution. Formally, for all $t \in \mathbb{N}$ and any $\epsilon > 0$ it holds that: $\|p_\theta(\boldsymbol{x}_t \mid \boldsymbol{m}_{[0,t]}) - p(\boldsymbol{x}_t \mid \boldsymbol{m}_{[0,t]})\| < \epsilon$.*

**Parallel-in-time models are on-trajectory under Assumption 1.** We derive this in Appendix B.1. Intuitively, for each $\epsilon$ we can choose the window $h$ large enough to guarantee arbitrary closeness. In contrast, autoregressive models are in general not on-trajectory. As a counterexample, consider an underlying chaotic system where the model ignores the measurements (for further discussion, see Hess et al. (2023) and Appendix B.2).

**Over-reliance on the autoregressive state.** The above analysis highlights a fundamental problem we call over-reliance on the autoregressive state. Intuitively, it states that an autoregressive state plays an ambiguous role. If the state is on-trajectory, it helps the model as it provides useful information for the next state. If the state is off-trajectory, the model may still make use of it for prediction and drift off further. In principle, the only way for the model to correct an off-trajectory autoregressive state is with informative measurements. This excludes edge cases, e.g., where the model converges to a global steady state.

Ideally, a model would detect when the autoregressive state drifts off and decrease its influence on the prediction. Depending on the informativeness of the measurements, it could increase its reliance on measurement information or capture the inherent uncertainty in a different way.

Over-reliance on autoregressive state is also influenced by the training strategy. Autoregressive models are usually trained with teacher-forcing, where the ground-truth previous state $x_t$ is used to predict the next state $x_{t+1}$. As these states are highly correlated, the model might learn to over-rely on $x_t$ and under-rely on $m_{t+1}$. This could potentially be mitigated by training techniques such as generalized teacher forcing (Hess et al., 2023) or unrolled training (Kohl et al., 2023).

How much information is taken from the autoregressive state is indicated by $J_t(x_t)$. To avoid autoregressive error accumulation, the Jacobian product series should ideally have a small norm. Notice that parallel-in-time models predict $x_{[t-h,t]} = f_\theta(m_{[k,t]}, \eta)$ without any recurrence. Consequently the Jacobians are all zero-matrices.

> **Advice for practitioners**
>
> The selection of an *on-trajectory* neural twin approach depends on two key factors:
> **(1) Measurement Informativeness.** Highly informative measurements often justify the use of autoregressive models (e.g., Kalman filters). If measurements lack sufficient information for state reconstruction, *parallel-in-time models* are preferable to ensure staying on-trajectory – **but** they require the selected window's measurements to be informative.
> **(2) System Dynamics.** If measurements provide limited information, chaotic systems typically demand *parallel-in-time models* to stay on-trajectory. For stable or periodic systems (e.g., steady-state behavior) *autoregressive models* may suffice.

## 4 EXPERIMENTS

### 4.1 MODELS

**PAINT via Flow Matching.** We implement FlowPAINT as a concrete instantiation of PAINT using Flow Matching (Albergo & Vanden-Eijnden, 2022; Albergo et al., 2023a; Lipman et al., 2022; Liu et al., 2022). PAINT is agnostic to the neural network architecture. We therefore aimed for simplicity and use established building blocks of the computer vision literature (Dosovitskiy et al., 2020; Hoogeboom et al., 2023; 2024; Peebles & Xie, 2023). Further experimental details can be found in Appendix A.2.

**Data-coupled stochastic interpolants.** PAINT is also agnostic to the generative modeling paradigm, as long as it maps a source to a target distribution. We follow the modeling from Albergo et al. (2023b) and model the unmasking as data-coupled distribution matching. However, the masking ratio of our method is very high (25 of 16K pixels are unmasked). In early experiments we observed that the gradient signal was too weak for the vicinity of the probes. Therefore, we introduce a spatially weighted loss that puts higher weight on pixels in the neighborhood of the probe points. As stochastic interpolants and Flow Matching in principle lead to the same training setup, we use a common Flow Matching implementation (Lipman et al., 2024).

**Autoregressive UNet.** To compare PAINT to an autoregressive model, we use the UNet architecture from Kohl et al. (2023). We follow their proposed paradigm and condition on the probe values

$m_{t+1}$ by concatenating the mask and probe values to the channels. The model sizes are chosen to match parameters. FlowPAINT has 19.8M and the autoregressive UNet 20.6M. Further architectural details and hyperparameters are reported in Appendix A.2.

## 4.2 DATASET AND EVALUATION

**Turbulent single jet dataset.** We generate a CFD dataset using the incompressible, pressure-based solver pimpleFOAM in OpenFOAM 8 (OpenCFD, 2009), with subgrid scales modeled by the Smagorinsky LES model (Pope, 2001; Fröhlich, 2006). Simulations are performed on a structured mesh, and for training purposes, a subregion is extracted and interpolated onto a regular grid. Since the state-space represents 2D velocities, we will write $\boldsymbol{u}_t$ instead of $\boldsymbol{x}_t$ with $\boldsymbol{u}_t \in \mathbb{R}^{d_1 \times d_2}$. The train/val/test splits across Reynolds numbers are provided in Appendix A.1.

**Physical coherence.** Due to the chaotic and rapidly diverging nature of turbulent flows, comparing a single predicted frame or a short sequence to the ground truth is not meaningful. Research in turbulence instead focuses on the flow's inherently reproducible statistical properties (Pope, 2001; Davidson, 2015). Given a possibly infinitely long trajectory $\boldsymbol{u}_{[0,t]}$, we follow a common notation in fluid dynamics and decompose $\boldsymbol{u}_{[0,t]}$ into a *time-averaged mean* component $\overline{\boldsymbol{u}} := \lim_{t\to\infty} \frac{1}{t} \sum_{i=1}^{t} \boldsymbol{u}_i$ and a turbulent fluctuation component $\boldsymbol{u}'_{[0,t]}$, such that $\boldsymbol{u}_{[0,t]} = \overline{\boldsymbol{u}} + \boldsymbol{u}'_{[0,t]}$. The *variance over time* is denoted as $\overline{\boldsymbol{u}'^2} := \lim_{t\to\infty} \frac{1}{t} \sum_{i=1}^{t} (\boldsymbol{u}_i - \overline{\boldsymbol{u}})^2$. Further, we define $E(k)$ as the *mean kinetic energy spectrum*. While these metrics are defined over an infinite-time, in practice they are computed over a large, finite horizon.

**Probe constellations.** The model was trained with 25 randomly sampled probe points for each data point in a batch. At inference, we investigate the two probe point constellations depicted in Figure 3. *Grid*: A 10×10 grid of 100 probe points evenly spaced across the domain (see Figure 3a). *Vertical*: A linear arrangement of 25 probe points positioned at the 3/4 axial location of the domain (see Figures 3b). In both configurations, an additional probe point is placed at the center of the inlet of the jet to capture the inlet velocity.

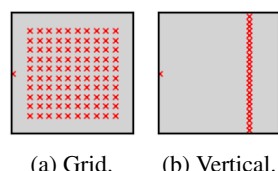

(a) Grid.    (b) Vertical.

Figure 3: Probe positions.

## 4.3 RESULTS

Table 1: Results for physical coherence of FlowPAINT (ours) and the autoregressive UNet baseline (AR UNet). The MAE, MSE and RMSE are taken between the ground truth and the predicted quantity. $\overline{\mathbf{u}}$ and $\overline{\mathbf{u}'^2}$ denote the mean and variance over time. $E(k)$ is the predicted kinetic energy spectra. $\mathrm{MSE}(u_{[0,t]})$ is the mean squared error between the predicted and the ground-truth trajectory as a whole. All values are taken along 900 timesteps in the test trajectories. The values after $\pm$ are standard deviations from averaging over 10 seeds. Generally, the higher Reynolds number seems to be more difficult for both models.

| Model | $Re$ | Probes | $\mathrm{MAE}(\overline{\mathbf{u}}) \downarrow$ | $\mathrm{MAE}(\overline{\mathbf{u}'^2}) \downarrow$ | $\mathrm{MSE}([\mathbf{u}_{[0,t]}]) \downarrow$ | $\mathrm{RMSE}(E(k)) \downarrow$ |
|---|---|---|---|---|---|---|
| | | | $\times 10^{-3}$ | $\times 10^{-5}$ | $\times 10^{-5}$ | $\times 10^{-6}$ |
| FlowPAINT | 2100 | grid | $1.2 \pm 1.6$ | $9.9 \pm 21.7$ | $8.1 \pm 3.6$ | 6.85 |
| AR UNet | 2100 | grid | $9.5 \pm 14.3$ | $98.7 \pm 110.4$ | $158.3 \pm 46.8$ | 70.9 |
| FlowPAINT | 2100 | vertical | $2.2 \pm 3.1$ | $30.2 \pm 38.4$ | $49 \pm 21$ | 20.1 |
| AR UNet | 2100 | vertical | $9.2 \pm 13.8$ | $97.8 \pm 108.8$ | $157.3 \pm 70.0$ | 71.8 |
| FlowPAINT | 1100 | grid | $0.9 \pm 1.3$ | $2.2 \pm 5.5$ | $2 \pm 1$ | 1.98 |
| AR UNet | 1100 | grid | $3.5 \pm 5.6$ | $17.3 \pm 25.9$ | $28.8 \pm 8.3$ | 12.4 |
| FlowPAINT | 1100 | vertical | $1.1 \pm 1.6$ | $6.3 \pm 11.1$ | $8.9 \pm 3.4$ | 3.68 |
| AR UNet | 1100 | vertical | $3.4 \pm 5.5$ | $16.9 \pm 25.5$ | $28.8 \pm 8.1$ | 12.2 |

**Quantitative results.** The quantitative results are reported in Table 1. FlowPAINT outperforms the autoregressive UNet across all metrics, probe constellations and Reynolds numbers in the test set. We conclude that FlowPAINT is more physically coherent than the probe-conditioned autoregressive baseline. Further results and analyses can be found in Appendix C.

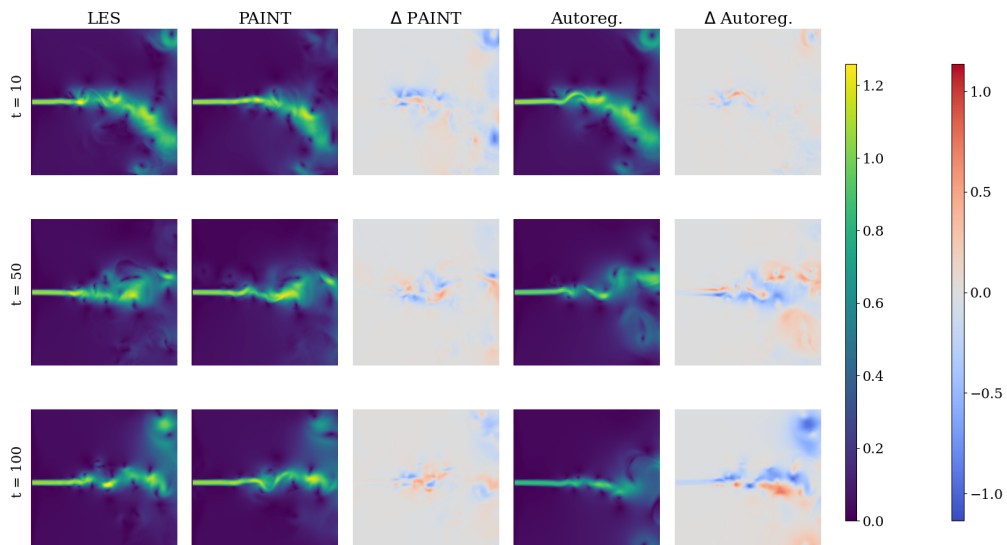

Figure 4: Ground truth vs. predicted states at different timesteps for $Re = 2100$ using the *vertical* probe constellation. PAINT (ours) does not accumulate errors over time. For the autoregressive UNet a pronounced drift over time is observable. All values are normalized by the mean inlet velocity.

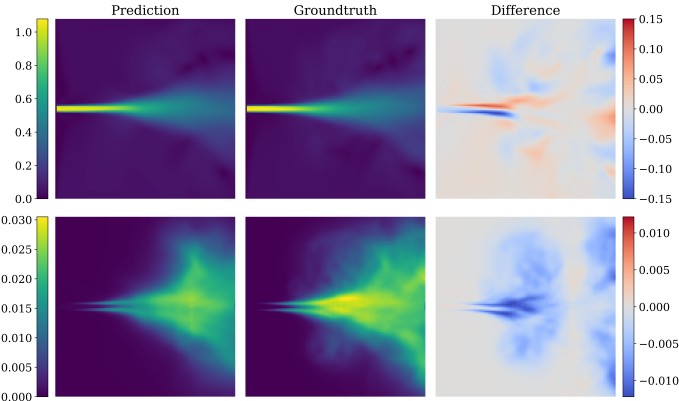

Figure 5: Mean velocity $\overline{u}$ and variance $\overline{u'^2}$ of the reconstructed $Re = 2100$ test trajectory using the *vertical* probe constellation compared with the ground truth. All values are normalized by the mean inlet velocity.

**Recovery of the physical flow characteristics.** We observe the models ability to reconstruct physical plausible states from different constellations of probe points (see Figures 5 and 6). The model's ability to reproduce the correct physical statistics seems to depend directly on the number and position of the sample points (as can be seen when comparing Figure 5 with Figures 10a, 11a and 12a ).

**Parallel-in-time model maintains stable error, while autoregressive baseline drifts.** The autoregressive approach exhibits a drift in MSE when rolled out (compare Figures 4, 7, 13,14 and 15). Importantly, the autoregressive model benefits from a known, correct initial state, which is often unknown in practice.

**Sampling a sequence vs. a single state.** PAINT can be used in two variants. The first samples a connected sequence, usually $\boldsymbol{x}_{[t-h,t]} \sim p(\boldsymbol{x}_{[t-h,t])} \mid \boldsymbol{m}_{[t-h,t]}$ (see Figure 16). This sequence is smooth over time and can be used by practitioners to interpret results.
The second variant samples a single timestep. It is used for state estimation in a sliding window

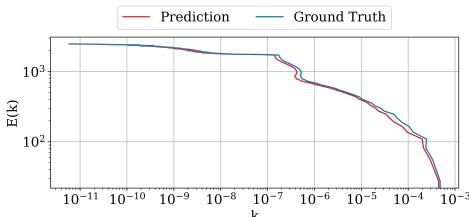

(a) Time-averaged kinetic energy spectrum.

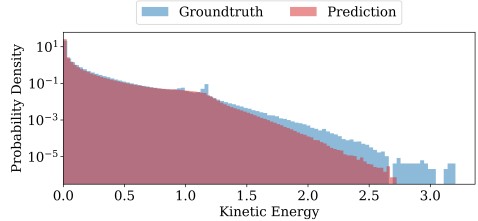

(b) Histogram of the kinetic energy.

Figure 6: Statistical analysis of reconstructed flow fields for Reynolds number 2100 and a *vertical* probe point constellation.

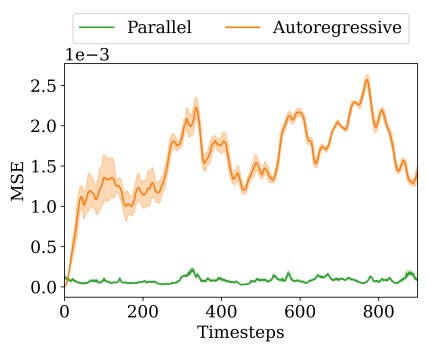

(b) *Grid* probe constellation.

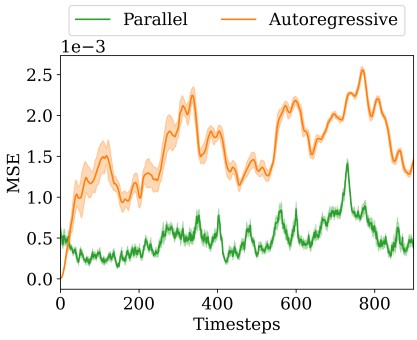

(a) *Vertical* probe constellation.

Figure 7: MSE over time of ground-truth vs. predicted trajectories for the $Re = 2100$ test trajectory and different probe point constellations. Interestingly, the MSE's of the two autoregressive trajectories are very similar, even if more informative probes are provided for the grid. The autoregressive model indicates over-reliance on the autoregressive state.

fashion by sampling $\boldsymbol{x}_t \sim p(\boldsymbol{x}_t \mid \boldsymbol{m}_{[t-h,t]})$. In this case a sampled sequence of consecutive states is not smooth over time.

**Measure of uncertainty.** One can obtain a measure for uncertainty by sampling several times and computing the variance. In Figure 9 it is visible that the model is most certain around the probe positions and most uncertain to the left and the right.

**Compute at training and inference.** FlowPAINT needs substantially more training resources and is slower at inference than the autoregressive baseline. It was trained on 12 A100 (64 GB VRAM) for 30 hours, whereas the UNet was trained on a single H100 (80GB VRAM) for 24 hours. Notably, the UNet takes about 66 milliseconds and FlowPAINT around 6.6 seconds for 20 denoising steps. Further details can be found in Appendix A.2.

## 5 CONCLUSION

In this paper we introduced parallel-in-time neural twins (PAINT) a novel data-driven method for reconstructing dynamical systems from real-time measurements. PAINT employs a generative neural network to model the joint conditional probability of system states. Our method provable stays on-trajectory and does not rely on an initial condition. The empirical validation with FlowPAINT on 2D turbulent fluid dynamics confirms its effectiveness. This paper represents an advancement in designing interpretable and widely applicable neural twins for complex dynamical systems.

**Limitations and future work** The biggest limitation of our method are the high computational costs, which deserves further investigation. Another limitation is that the sliding window approach does not generate continuous generations. This could potentially be mitigated by stitching techniques (Wei et al., 2019).

**Ethics statement.** More broadly, our work aims to advance the field of machine learning and may contribute to its broader societal impact. More specifically, our work advances reconstruction of dynamical systems, with a wide range of potential applications. While our method or future derivatives of it might enable transformative societal benefits it also introduces risks of misconduct or dual-use risks. To mitigate the risk of misconduct, we will strive to be transparent about the capabilities and limitations of this method and encourage the use in real-world applications only after rigorous testing. We welcome critical discussions on further safeguarding such technologies against adversarial use.

**Reproducibility statement.** We provide code to reproduce the main results in the supplementary material. Additionally, we report hyperparameters, and important implementation details to facilitate the reproduction of our results. Upon publication, the code will be made publicly available on GitHub.

**Usage of LLMs.** LLMs assisted in ideation and in refining and optimizing formulations on a sentence- and paragraph-level.

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

# A EXPERIMENTAL DETAILS

## A.1 DATASET

**Turbulent single jet dataset.** We use an in-house dataset to ensure its quality and match the specific requirements of our study. As we have the capability to generate reliable high-fidelity data, additional external datasets were not needed. We consider the domain shown in Figure 8, which produces a two-dimensional turbulent free jet. The inlet is prescribed by a turbulent power-law velocity profile, with the system dynamics controlled by the mean inlet velocity. The Reynolds number is defined with respect to the inlet velocity as

$$Re = \langle u \rangle_{\text{inlet}} h_{\text{inlet}} \nu^{-1}. \tag{7}$$

The dataset is generated using the incompressible, pressure-based solver `pimpleFOAM` in `OpenFOAM 8` (OpenCFD (2009)), with subgrid scales modeled by the Smagorinsky LES model (Pope, 2001; Fröhlich, 2006). Simulations are performed on a structured mesh consisting of 19040 cells, and for training purposes, a subregion is extracted and interpolated onto a regular $128 \times 128$ grid to facilitate adaptability.. The numerical simulations made use of a timestep size of $\Delta t = 6.5 \times 10^{-4}$s. For training purposes, the data are sampled every 70th timestep, resulting in an effective temporal resolution of $4.55 \times 10^{-2}$s. For each Reynolds number, the dataset contains 1000 such snapshots.

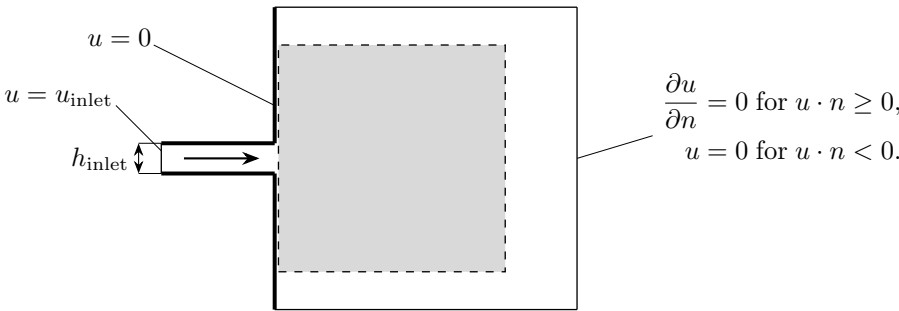

Figure 8: Computational domain of the CFD simulation, featuring a thin channel with a prescribed inlet velocity profile, transitioning into an expanded outflow region zone. The channel and outflow orifice walls enforce a no-slip boundary condition, while the top, bottom and right boundaries of the outflow region implement a conditional Neumann/Dirichlet outflow condition. The light-grey subregion denotes the part of the CFD domain considered for model training. The arrow shows the imposed inlet flow direction.

**A note on 2D turbulence.** In fluid mechanics, the behavior of turbulent flows differs fundamentally between three and two dimensions. In three dimensions, turbulence is characterized by the *energy cascade*: the largest eddies form due to inertial instabilities but tend to exist only briefly, breaking up into progressively smaller eddies. This cascade continues until viscous forces dissipate the smallest scales.

In contrast, in two dimensions, the energy cascade is reversed, transferring energy from smaller to larger eddies, which produces coherent, long-lived vortices. Intuitively, this can be explained by *vortex stretching*. Consider a thin tube of vorticity: if a velocity gradient along the tube axis is present, the tube stretches, and due to conservation of angular momentum, the vorticity magnitude increases. This mechanism is absent in two dimensions because there is no third dimension along which stretching can occur. As a result, vortices cannot break up and tend to merge into larger structures if they have the same rotational sense, which explains the inverse energy cascade.

Mathematically, this is seen in the vorticity transport equation derived from the curl of the Navier–Stokes equations. In 2D, the velocity and vorticity vectors are perpendicular, which makes the vortex stretching term identically zero.

In reality, there is no perfectly two-dimensional turbulence; even quasi-2D flows, such as atmospheric flows, exhibit three-dimensional structures at smaller scales. Nevertheless, we are confident that the proposed method, having demonstrated robust performance in two-dimensional turbulence,

will generalize effectively to three-dimensional turbulent flows. For a more detailed discussion of two-dimensional turbulence, see Davidson (2015).

**Train/val/test splits.** We consider 18 different training trajectories with varying Reynolds numbers, as explained above. The Reynolds numbers used range from 700 to 2400 in steps of 100. The split is performed at random, but avoiding that any validation or test data is placed in the extrapolation regime. The exact split can be seen in Table 2.

Table 2: Reynolds numbers by split.

| Reynolds | train | val | test |
|---|---|---|---|
| 700 | X | | |
| 800 | X | | |
| 900 | X | | |
| 1000 | X | | |
| 1100 | | | X |
| 1200 | X | | |
| 1300 | X | | |
| 1400 | X | | |
| 1500 | X | | |
| 1600 | | X | |
| 1700 | X | | |
| 1800 | X | | |
| 1900 | X | | |
| 2000 | X | | |
| 2100 | | | X |
| 2200 | X | | |
| 2300 | X | | |
| 2400 | X | | |

### A.2  IMPLEMENTATION AND ARCHITECTURE

**Architecture and hyperparameters.** FlowPAINT and the autoregressive UNet were both implemented in Pytorch (Paszke et al., 2017). For FlowPAINT we trained in float16 mixed precision. The most important training parameters are provided in Table 3. Our code also uses FlashAttention (Dao et al., 2022; Dao, 2024). For both models we use 20 denoising steps for generating samples.

**Autoregressive UNet.** For the UNet we followed (Kohl et al., 2023) and took the implementation of the public Github repository [2]. We matched the number of parameters by increasing the default model dimension from 128 to 224. For probe-conditioning, we followed the same recipe as for prior timestep conditioning and added mask and probe values as additional channels.

**Compute requirements.** FlowPaint is expensive during training, as it needs to reconstruct a video in parallel. We had limited access to a cluster environment and trained FlowPaint on 3 nodes each consisting of 4 NVIDIA Ampere A100 GPUs with 64GB VRAM. Training took around 30 hours for 100K iterations.
The autoregressive baseline is comparably cheap. It was trained on a single Nvidia H100-SXM-80GB for ca. 24 hours. The inference times are reported in the main paper.

## B  THEORY

### B.1  PARALLEL-IN-TIME MODELS ARE ON-TRAJECTORY

Here we provide a simple derivation to show that parallel-in-time models are on-trajectory under Assumption 1. We will show that for an appropriate window size $h$, for all $t \in \mathbb{N}$ and any $\epsilon_1 > 0$:

$$||p_\theta(\boldsymbol{x}_t \mid \boldsymbol{m}_{[t-h,t]}) - p(\boldsymbol{x}_t \mid \boldsymbol{m}_{[0,t]})|| < \epsilon_1. \qquad (8)$$

---

[2]https://github.com/tum-pbs/autoreg-pde-diffusion

Table 3: Experimental details for FlowPaint and the autoregressive baseline.

|  |  | FlowPaint | AR baseline |
|---|---|---|---|
| Data |  |  |  |
|  | History length | 16 | 2 |
|  | Forward prediction | 8 | 1 |
| Optimization |  |  |  |
|  | Optimizer | AdamW | AdamW |
|  | AdamW decay | 0.05 | 0.05 |
|  | Learning rate | $10^{-4}$ | $10^{-4}$ |
|  | Learning rate start | $5 \cdot 10^{-7}$ | $5 \cdot 10^{-7}$ |
|  | Learning rate warmup steps | 10000 | 10000 |
|  | Learning rate end | $10^{-5}$ | $10^{-5}$ |
|  | Train steps | 100K | 100K |
|  | Batch size | 144 | 144 |

Table 4: Architectural details of FlowPaint.

| Encoding |  |  |
|---|---|---|
|  | num_layers of 3x3x3 conv | 3 |
|  | patch size in pixels | 4x4 |
| Transformer |  |  |
|  | model_dim | 192 |
|  | layers | 10 |
|  | num_heads | 3 |
|  | Spatial Block | Yes |
|  | Temporal Block | Yes |
|  | Temporal Convolution | No |
| Decoding |  |  |
|  | num_layers of 3x3x3 conv | 3 |

First, from Assumption 1 choose a window size $h$ such that:

$$||p(\boldsymbol{x}_t \mid \boldsymbol{m}_{[t-h,t]}) - p(\boldsymbol{x}_t \mid \boldsymbol{m}_{[0,t]})|| < \epsilon_2 < \frac{\epsilon_1}{2}. \tag{9}$$

Then, from the Universal Function Approximator Theorem (Cybenko, 1989), choose an approximation of the true distribution as follows:

$$||p_\theta(\boldsymbol{x}_t \mid \boldsymbol{m}_{[t-h,t]}) - p(\boldsymbol{x}_t \mid \boldsymbol{m}_{[t-h,t]})|| < \epsilon_3 < \frac{\epsilon_1}{2}. \tag{10}$$

Summing up the individual $\epsilon_1$ and $\epsilon_2$ the total approximation error is: $\epsilon_2 + \epsilon_3 < \frac{\epsilon_1}{2} + \frac{\epsilon_1}{2} < \epsilon_1$.

## B.2 IN GENERAL, AUTOREGRESSIVE MODELS ARE NOT ON-TRAJECTORY

Here we show a more detailed counterexample to illustrate that autoregressive model of the form

$$p(\boldsymbol{x}_{[0,t]} \mid \boldsymbol{m}_{[0,t]}) = p(\boldsymbol{x}_0 \mid \boldsymbol{m}_{[0,t]}) \cdot \prod_{i=1}^{t} p_\theta(\boldsymbol{x}_i \mid \boldsymbol{x}_{i-1}, \boldsymbol{m}_{[0,t]}), \tag{11}$$

do not generally fulfill the on-trajectory property. We construct this example while demonstrating the over-reliance on autoregressive state. Being on-trajectory is defined as: for all $t \in \mathbb{N}$ and any $\epsilon > 0$:

$$||p_\theta(\boldsymbol{x}_t \mid \boldsymbol{m}_{[0,t]}) - p(\boldsymbol{x}_t \mid \boldsymbol{m}_{[0,t]})|| < \epsilon. \tag{12}$$

From Section 3.3 it is known that autoregressive models accumulate errors with the product of Jacobians, similar to classical ODEs (Orrell, 2005).

**Logistic map.** Let $f : \mathbb{R} \mapsto \mathbb{R}$ be the logistic map, which is chaotic for $r = 3.8$ (May, 1976):

$$f(x_{t-1}, \eta_t) = r \cdot x_{t-1} \cdot (1 - x_{t-1}) \tag{13}$$

Then we assume the model $f_\theta$ ignores the measurements and learns to predict $x_t$ from $x_{t-1}$ alone, but with a small error $\epsilon' > 0$.

$$f_\theta(x_{t-1}, m_{[0,t]}, \eta_t) = r \cdot x_{t-1} \cdot (1 - x_{t-1}) + \epsilon' \tag{14}$$

This is an extreme case of over-reliance on autoregressive state, namely solely relying on it and not relying on the measurements at all. For infinitesimal perturbations in $x_{t-1}$ exponentiate over time, causing $p_\theta(x_t \mid m_{[0,t]})$ to diverge from the true trajectory distribution $p(x_t \mid m_{[0,t]})$ even for arbitrarily small $\epsilon'$. Thus, the autoregressive factorization is not on-trajectory because the model's sequential predictions amplify initial errors, violating the required closeness condition for all $t > 0$.

## C    ADDITIONAL RESULTS

We show additional results for other Reynolds numbers and probe constellations in Figures 9, 10, 11, 12, 14, 13, 15, and 16.

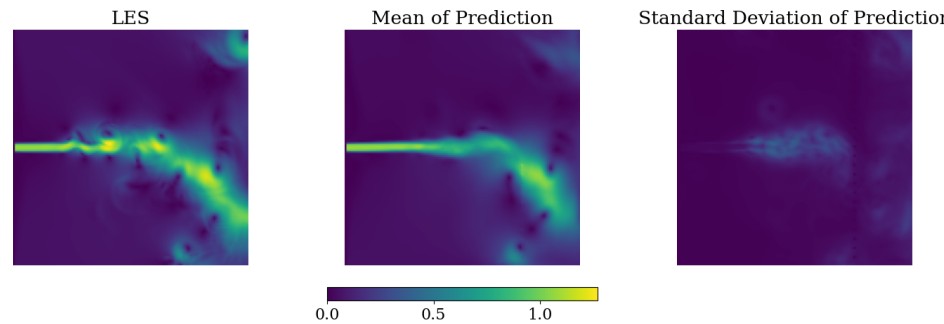

Figure 9: Left: Exemplary groundtruth velocity snapshot, for Reynolds number 2100 and using the *vertical* probe point constellation. Center: Mean of the predictions over 10 independent seeds. Right: Standard deviation of the predictions over the same 10 seeds. All values are normalized by the mean inlet velocity.

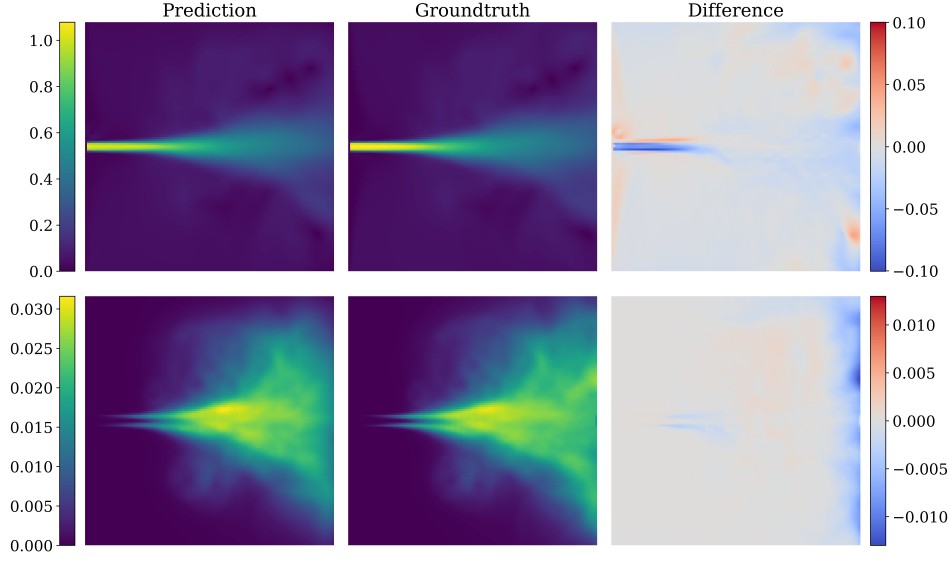

(a) Mean velocity ($\overline{u}$) and variance ($\overline{u'^2}$) of the reconstructed trajectories compared with the ground truth. All values are normalized by the mean inlet velocity.

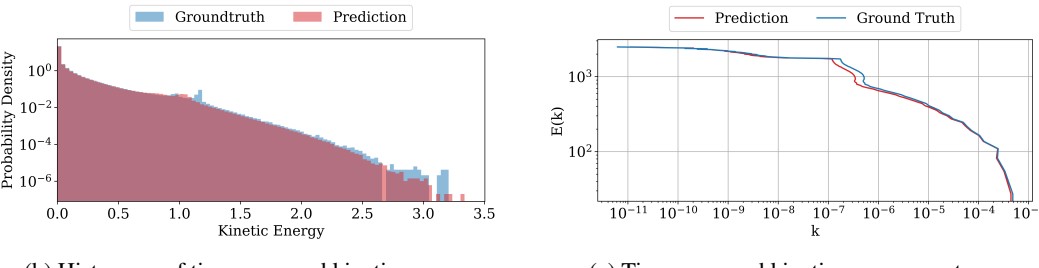

(b) Histogram of time-averaged kinetic energy.   (c) Time-averaged kinetic energy spectrum.

Figure 10: Statistical analysis of reconstructed flow fields for Reynolds number 2100 and a *grid* probe point constellation.

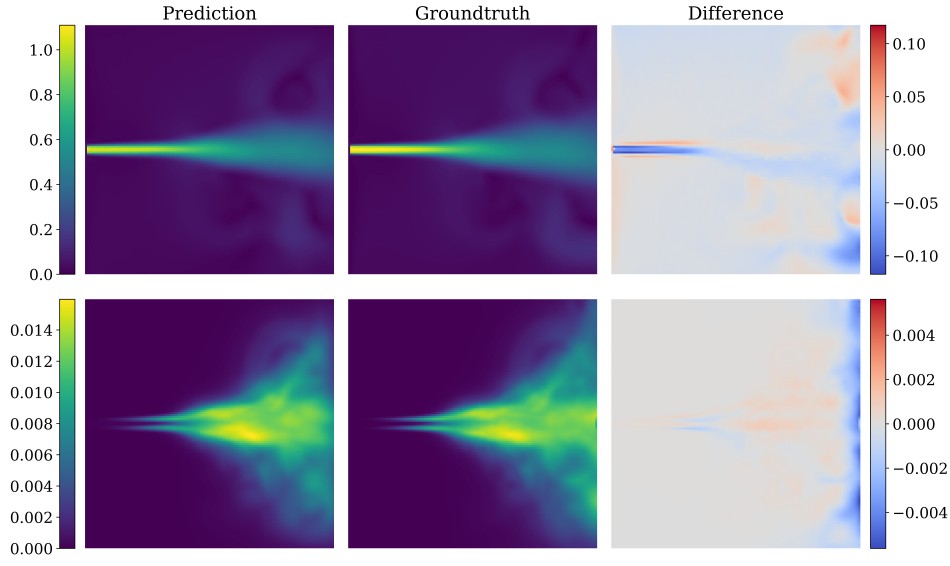

(a) Mean velocity ($\overline{u}$) and variance ($\overline{u'^2}$) of the reconstructed trajectories compared with the ground truth. All values are normalized by the mean inlet velocity.

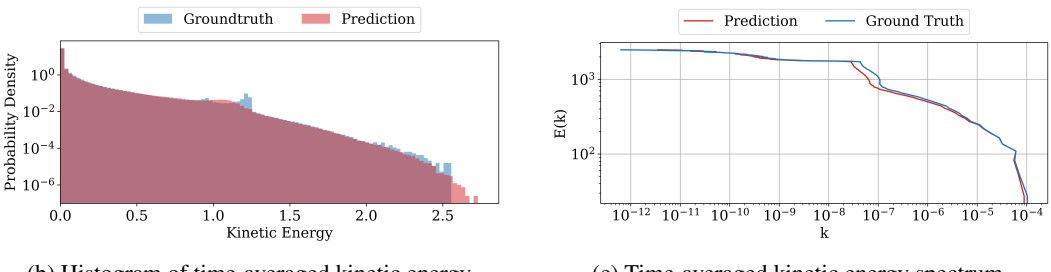

(b) Histogram of time-averaged kinetic energy.

(c) Time-averaged kinetic energy spectrum.

Figure 11: Statistical analysis of reconstructed flow fields for Reynolds number 1100 and a *grid* probe point constellation.

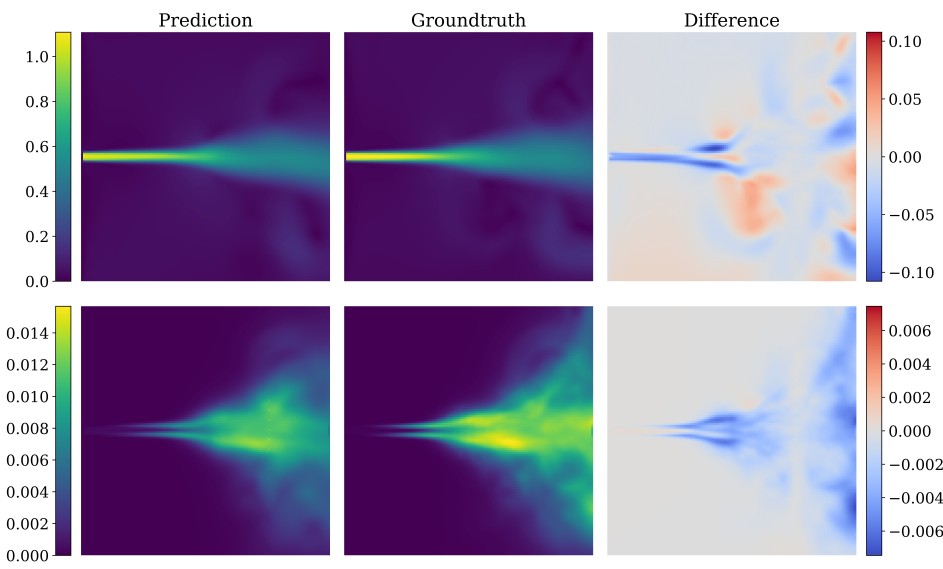

(a) Mean velocity ($\overline{u}$) and variance ($\overline{u'^2}$) of the reconstructed trajectories compared with the ground truth. All values are normalized by the mean inlet velocity.

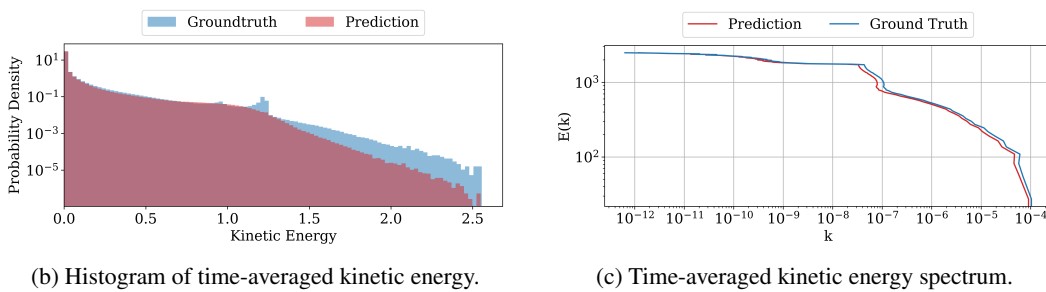

(b) Histogram of time-averaged kinetic energy.     (c) Time-averaged kinetic energy spectrum.

Figure 12: Statistical analysis of reconstructed flow fields for Reynolds number 1100 and a *vertical* probe point constellation.

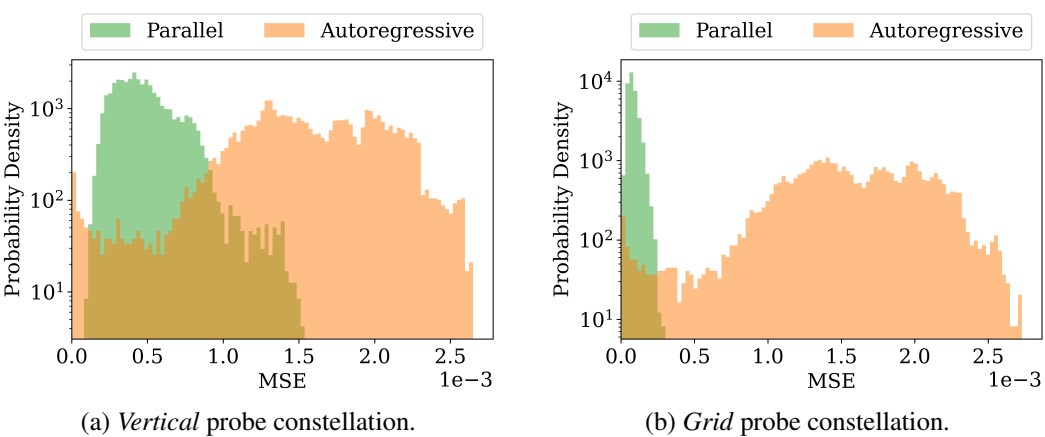

(a) *Vertical* probe constellation.            (b) *Grid* probe constellation.

Figure 13: Histogram of all MSE values of the Reynolds number 2100 test trajectory and a *grid* probe point constellation.

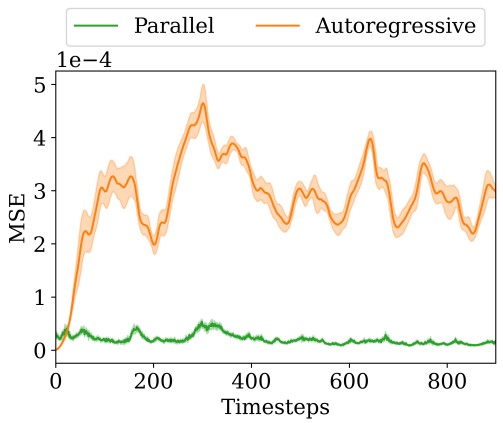

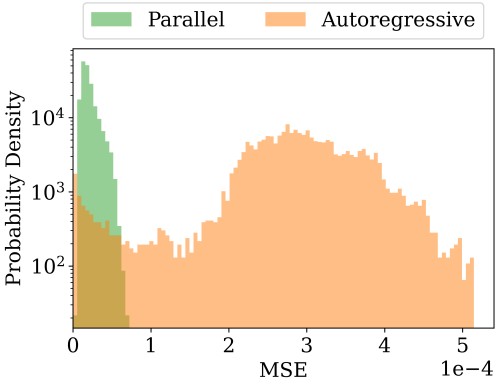

(a) Temporal evolution of the mean and standard deviation of the MSE when comparing the parallel-in-time and autoregressive approaches.

(b) Histogram of all MSE values accumulated over the rollout, comparing the parallel-in-time and autoregressive approaches.

Figure 14: Comparison of error characteristics between the parallel-in-time and autoregressive reconstruction approaches for Reynolds number 1100 and a *grid* probe point constellation.

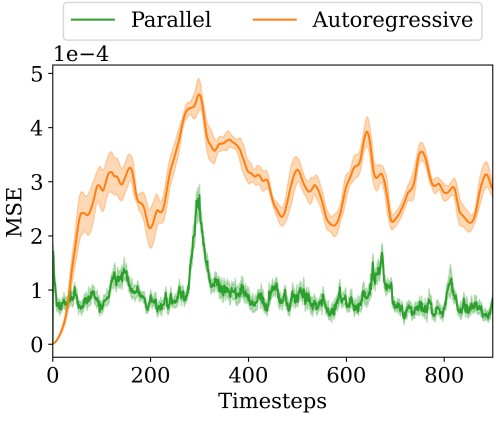

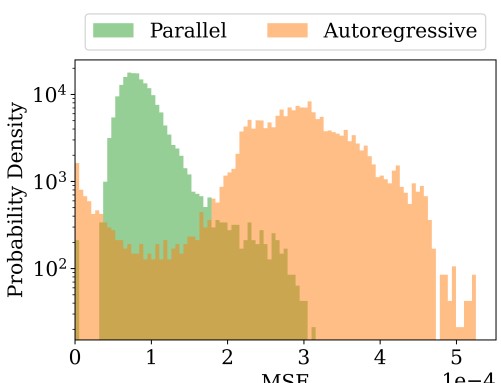

(a) Temporal evolution of the mean and standard deviation of the MSE when comparing the parallel-in-time and autoregressive approaches.

(b) Histogram of all MSE values accumulated over the rollout, comparing the parallel-in-time and autoregressive approaches.

Figure 15: Comparison of error characteristics between the parallel-in-time and autoregressive reconstruction approaches for Reynolds number 1100 and a *vertical* probe point constellation.

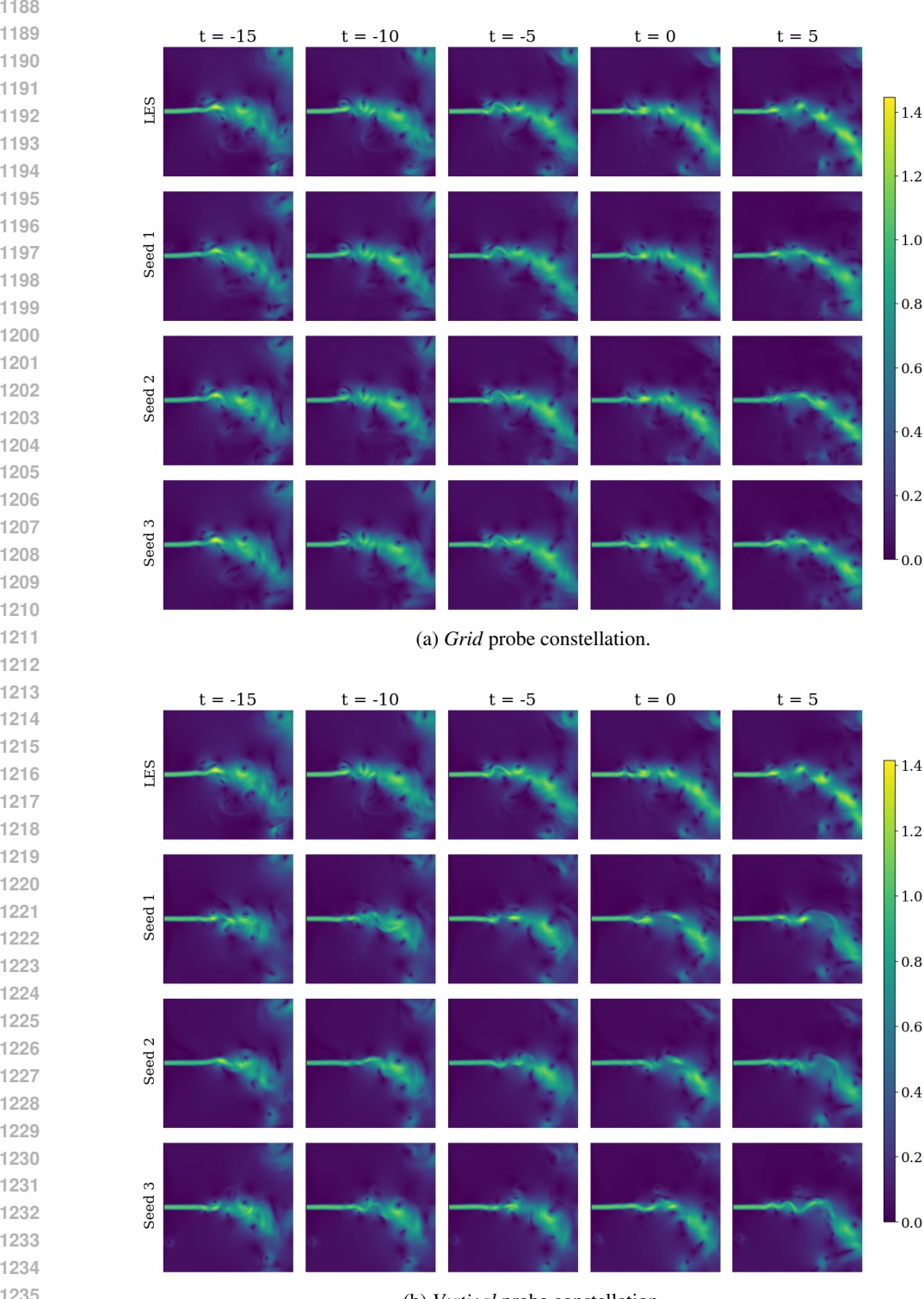

Figure 16: The top rows show five exemplary LES ground truth snapshots. Time steps with $t \leq 0$ correspond to reconstructions based on measurements, while time steps with $t > 0$ represent predictions with no measurements present. The following three rows display reconstructions from three independent random seeds, each showing a connected sequence generated based on the probe information provided. All values are normalized by the mean inlet velocity.

