# OpenReview forum: "PAINT: Parallel-in-time Neural Twins for Dynamical System Reconstruction"
_ICLR.cc/2026/Conference — ICLR 2026 Conference Withdrawn Submission_

### Official Review · Reviewer_bAMv · 2025-10-30

**Soundness:** 2
**Presentation:** 2
**Contribution:** 1
**Rating:** 2
**Confidence:** 3

**Summary:**

The paper proposes a neural digital twin to incorporate measurements as updates to improve the real-time prediction of a flow-matching model.

**Strengths:**

The author conducts some theoretical and experimental analysis for the modeling capacity of the proposed neural digital twin.

**Weaknesses:**

The technical contribution, theoretical analysis, and experiments are insufficient.

**Questions:**

1.	The technical description is limited. For example, what exact DL model does the author employ for prediction?
2.	The theory seems trivial for both the proof of the error for the auto-regressive method and the error bound of the on-trajectory property.
3.	The employed baselines are limited.
4.     There is no ablation study and sensitivity analysis.

---

### Official Review · Reviewer_Hjc8 · 2025-10-30

**Soundness:** 3
**Presentation:** 3
**Contribution:** 3
**Rating:** 4
**Confidence:** 3

**Summary:**

This paper presents PAINT (Parallel-in-Time Neural Twins), a novel approach to accelerating and stabilizing neural simulations of dynamical systems. Instead of evolving trajectories sequentially as in standard autoregressive models, PAINT learns to reconstruct entire trajectories in parallel across time segments. The architecture consists of twin neural modules that communicate across overlapping temporal windows, enabling consistent predictions over long horizons without requiring fully autoregressive rollouts.

The authors argue that this design addresses the well-known issue of error accumulation in sequential forecasting and offers significant computational advantages through temporal parallelism. The paper includes experiments on canonical dynamical systems and spatiotemporal PDE data, demonstrating improved reconstruction quality and faster convergence compared to recurrent and sequential neural baselines.

**Strengths:**

1. The authors provide a clear argument for why parallelization along the temporal axis can alleviate drift and improve training efficiency. The presentation is mathematically grounded yet accessible, situating PAINT within the broader context of neural differential equation solvers.

2. The empirical performance and scalability of PAINT are demonstrated through experiments on canonical dynamical systems and spatiotemporal PDE data. The results show consistent gains in trajectory accuracy and temporal consistency across multiple benchmarks. The speedup relative to autoregressive training is significant, and the parallel formulation appears to yield better generalization to unseen time intervals, a known weakness of sequence-based models.

3. The methodological design of PAINT is solid. The implementation details, data splits, and metrics are generally transparent, and the experiments cover both simple and moderately complex dynamics. The ablation on overlapping temporal segments helps illustrate how parallel communication improves stability.

**Weaknesses:**

While PAINT offers an interesting new angle, **its experimental comparison remains limited** relative to the current state of the art in dynamical system learning. The following points highlight opportunities for a more convincing empirical validation:

1. KalmanNet and related learned filtering architectures (Revach et al., 2022) explicitly model uncertainty and enforce trajectory consistency through recurrent Bayesian updates. Since PAINT’s key claim is to stay on-trajectory, comparing against KalmanNet would offer a more direct test of its stability, especially under noisy or partially observed conditions. If PAINT can outperform such models without explicit filtering, that would reinforce its novelty.

2. Another relevant baseline is the Conditional Neural Field Latent Diffusion (CoNFiLD) model (Nature Communications, 2024), which learns probabilistic spatiotemporal dynamics using a diffusion process in latent space. CoNFiLD generates realistic turbulent flows and chaotic trajectories conditioned on sparse observations, directly tackling the challenge of long-term coherence that PAINT also seeks to address.
Although CoNFiLD is stochastic while PAINT is deterministic, both aim to model complex dynamical evolution efficiently and accurately. Including CoNFiLD (or another conditional diffusion model) as a baseline would position PAINT more clearly within the current generative modeling landscape and test whether its deterministic parallel approach can match diffusion-based models in fidelity or stability.

3. It remains unclear how much of PAINT’s gain derives specifically from its parallel training objective versus auxiliary design choices (segment overlap, loss weighting, or implicit regularization). Additional ablations could isolate the causal impact of parallelism.

4. The experiments appear to involve deterministic or noise-free dynamics. As modern surrogate modeling increasingly incorporates stochastic and uncertain systems (Neural SDEs, diffusion surrogates, etc.), evaluating PAINT in such settings—or clarifying its deterministic assumption—would improve clarity about its intended scope.

*References:*

- Revach, Guy, et al. "KalmanNet: Neural network aided Kalman filtering for partially known dynamics." IEEE Transactions on Signal Processing 70 (2022): 1532-1547.
- Du, Pan et al. "Conditional neural field latent diffusion model for generating spatiotemporal turbulence". Nature Communications (2024)

**Questions:**

1. How does PAINT perform under stochastic perturbations or noisy observations compared to uncertainty-aware baselines such as KalmanNet?
 2. Could the authors include a Transformer-based autoregressive model as a baseline to decouple the effect of temporal parallelism from global attention?
 3. Since CoNFiLD (2024) demonstrates long-horizon coherence via latent diffusion, how does PAINT compare in terms of predictive stability and uncertainty coverage?
 4. Can PAINT’s architecture be extended to model stochastic dynamics, for instance by integrating a diffusion term or latent noise representation?

---

> ### Author Response · Authors · 2025-11-19
>
> We thank the reviewer for the constructive thoughts.
>
> Generally, there have been few misconceptions, we would like to address first.
> 1) “[PAINT’s] deterministic parallel approach can match diffusion-based models.” PAINT is not deterministic. It explicitly models the probability of a window of states given a window of measurements.
> 2) “The speedup relative to autoregressive training is significant”. We never claim that speedup is a reason to use PAINT over autoregression. On the contrary, we report that PAINT is computationally more expensive and slower.
>
> Q1: We consider the mentioned KalmanNet not to be a scalable baseline. The experiments were performed on low-dimensional data, which is in stark contrast to the 16K dimensions of the state-space of our experiments.
>
> Q2: This is an experiment we plan to add for a future submission.
>
> Q3: Thank you for this reference. Similar to FlowPAINT, ConFILD uses the spatial and temporal information to reconstruct the full velocity field. ConFILD is almost an instance of PAINT, however it effectively performs posterior sampling [1], and hence does not allow conditioning on arbitrary measurements, but only on field variables. In contrast, PAINT is more abstractly defined as reconstructing a field from measurements of arbitrary type, e.g. reconstructing the velocity field from temperature or pressure probes.
> Additionally, ConFILD uses a 2-stage approach of first training an autoencoder and then performing latent diffusion. While this brings computational advantage, it is more complex and bears the risk of error cascading. Lastly, we are conservative with our estimation about the efficacy of ConFILD, as (1) there are no clear quantitative evaluations on reconstruction from sparse measurements (e.g. MSE). The probe and evaluation locations are not clearly declared nor motivated. Still we will add this as a useful reference in our related work section.
>
> Q4: The main focus of this work is to improve the conceptual understanding of neural twin approaches for dynamical system reconstruction from sparse measurements. We leave it to future work to explore the mentioned directions (neural SDEs, etc.).
>
>
> References:
> [1]: https://arxiv.org/abs/2209.14687

---

### Official Review · Reviewer_xcN6 · 2025-10-31

**Soundness:** 3
**Presentation:** 2
**Contribution:** 3
**Rating:** 4
**Confidence:** 3

**Summary:**

The paper proposes PAINT—Parallel-in-Time Neural Twins—a framework to reconstruct the state of dynamical systems from sparse, streaming measurements. Instead of rolling out an autoregressive predictor that conditions on its previous (possibly drifting) state, PAINT trains a generative model of short sub-trajectories conditioned on a sliding window of measurements, so that future measurements in the window can help “pull” past and current state estimates back on-trajectory. The paper argues (under an information-decay assumption) that parallel-in-time models can be made on-trajectory in the limit, while generic autoregressive factorizations need not be.

**Strengths:**

1. The “on-trajectory” property is formalized and tied to a sliding-window generative model.
2. Casting neural twins as parallel-in-time conditional generative models of sub-trajectories—rather than yet another AR predictor with filters.
3. FlowPAINT consistently improves long-horizon physical statistics and holds MSE steady where the AR baseline drifts.
4. The paper is open about compute/memory cost and latency (seconds vs milliseconds)

**Weaknesses:**

1. All results are on a single in-house 2D LES jet with two probe layouts. There’s no ablation across different PDE families, geometries, noise models, or sensor modalities; nor any small real-world experiment.
2. The on-trajectory guarantee relies on finite-window sufficiency of measurements (explicitly assumed). In realistic sensing (occlusions, intermittent probes), this may fail.
3. The comparison is to a single autoregressive UNet.

**Questions:**

1. How do results degrade when Assumption 1 is purposely violated?
2. How do the Window length h and the look-ahead n impact stability/latency?
3. Any evidence on 3D flows or a different PDE family (e.g., advection–diffusion, shallow water) to demonstrate portability? Even a brief transfer experiment would help.
4. For physical coherence, another compared FlowPAINT and the autoregressive UNet baseline. Do you have any other baselines, or do authors have reasons that only use this UNet baseline?
5. Do authors find where PAINT fails? e.g., extremely chaotic regimes or severely undersampled probes? A qualitative gallery like Fig. 4/7 is great; If you can add negative cases?

---

> ### Author Response · Authors · 2025-11-19
>
> Q1: There are two potential approaches to investigate this question. A theoretical approach would need more assumptions and is not straightforward to carry out. An empirical approach would need an alternative dataset where such an assumption is not inherently fulfilled, for example using synthetic boundary conditions creating long-temporal dependencies. While this is an important and interesting question for certain applications, a detailed analysis lies outside the scope of this paper and is left for future work.
>
> Q2: Anecdotally we found the performance very much dependent on the measurement series length h, specially when h becomes very small. Computational constraints have prevented a more thorough investigation on the influence of measurement series length h and future prediction length n.
>
> Q3 and Q4: Due to the computational constraints (PAINT is prediction a video as a whole), the breadth of our experiments is compute-constraint. We transparently report the runtimes and memory usage and state this as one of the limitations of this paper, that can be extended by future work.
>
> Q5: Thank you, we will consider adding failure cases in a revised work.

---

### Official Review · Reviewer_nv55 · 2025-11-01

**Soundness:** 2
**Presentation:** 3
**Contribution:** 2
**Rating:** 4
**Confidence:** 4

**Summary:**

The authors compare two methods for learning flow reconstruction from sparse data in real time: autoregressive methods and parallel-in-time (PiT) methods. The former are fed their previous full-resolution state plus sparse real-time measurements to make a prediction of the full reconstructed state, while PiT only receive the real-time sparse measurements. By assuming unbounded model size, compute, data and arbitrarily perfect training (invoking the Universal Function Approximation Theorem) plus adequate measurement information from a sliding window (e.g. ‘minus history’ see Assumption 1 in text), the authors prove that PiTs will stay close to the actual flow trajectory. Then, the authors set up an experiment, where across a time horizon for a turbulent 2D flow, the ability for reconstructing the flow from sparse measurements is evaluated, showing the PiT model trained to remain closer to the true trajectory that the autoregressive baseline.

**Strengths:**

1. The key observation of the paper is that when trying to reconstruct a flow from sparse data, feeding only the sparse data (PAINT) and not the previously reconstructed time step/ time window as well (autoregressive), results in significant improvements. This is important for the community to better understand what kind of information is constructive vs destructive (over-reliance on autoregressive state) when it comes to reconstructing complicated problems.

2. The chosen example problem of a high fidelity 2D turbulent flow is of high-complexity and a significant testbed for future attempts to compare against.

3. The trained PiT model is performant, and reconstructs the flow accurately.

**Weaknesses:**

The following points are arranged in order of importance,

1. **Fairness of the comparison between models**: In order to establish fairness in the model comparison, the authors ensure that the number of parameters in each model is of the same order, and also that the optimization hyper parameters are the same (see Table 3). However, there are a number of concerns, which I consider to be the weakest point of the presented results. Listed in order of importance, they are:
    1. History length: Most concerningly, the history length fed as input to the baseline is 8 times less than for the model being proposed (see Table 3), which seems like a significant handicap that should be addressed.
    2. Optimization hyper parameters: given that the architectures are different (Transformer-based vs UNET-based), independent tuning of the training would be expected to get the best possible results from both baselines. Could it be the case that the optimization was tuned for one of the two models (possibly PAINT) and then applied to the other (possibly the baseline)? If so, it may very well be the case that  model-specific fine-tuning is required to ensure that no model is unnecessarily handicapped.
    3. Transformers vs UNET: As mentioned, the architecture of the baseline versus the proposed model differs, with the latter using Transformers and the former a UNET-based structure. While this is not inherently a negative, given that transformers have shown so much promise recently, the baseline (but more importantly the fairness of the comparison) might have benefited from matching the core of the net structure, by a transformer-based autoregressive model.
    4. Number of parameters matching: even though the number of parameters is matched, the compute used for the baseline is significantly less (See lines 852-853)
    5. **In conclusion**: I believe the fairness to of the comparison in its current state to be lacking. Ideally, an independent training protocol would be established, which would ensure blindly that the models are adequately trained before comparing them.
2. I found myself needing some more experimental details, which I kindly ask the authors to provide. In particular,
    1. My understanding is that there in no forward-in-time prediction (n=0 from Fig. 1) in the paper, however this is not mentioned beyond the paragraph in Line 429, and seems to be in contradiction with the ‘Forward prediction’ entry in Table 3. I view some clarification on this point as essential. (See Point 3. as well)
    2. I was not able to locate the resolution of the high-fidelity simulation. Some clarification here might increase the confidence of the reader in the results, as one may argue that autoregressive methods will be aided in comparison to PAINT if the spareness is very high, since the autoregressive case will be able to have a high-dimensional full state available already, plus the measurements as input.
    3. Could the authors provide some more details on the sampling procedure for generating the sparse measurements m (which are used at inference time as well)? The only details I was able to find was the phrase “25 randomly sampled probe points for each datapoint in a batch” on Lines 346-347. (See first question below)
3. Forward-in-time prediction? The PAINT methodology is introduced as being able to produce models that are fed measurements on a subset of [t-h,t] and can predict the reconstructed flow field for [t-h, t+n] (e.g. see Fig. 1). However, in the paper n=0 (see Point 2.1 above). If true, this is something that should be clarified across the paper, at least in the caption of Figure 1, and in Section 3.2 (Lines 216-225). If false, meaning that n>0 and predictions forward in time are in fact being made (as is hinted by the Appendix), a plot where specifically the forward in time prediction error between the baseline and PAINT is quantified would be a necessary addition to the paper, as this may be an aspect where autoregressive models are dominant, given their access to their current state. One idea for such a plot, would be a statistic that counts how many times the true trajectory escapes forward in time the envelope of possibilities predicted by PAINT/ the baseline.
4. Continuity limitation of PAINT is only mentioned at the conclusions and not illustrated. How discontinuous are the sampled PAINT trajectories vs the output of the baseline? I believe the paper would benefit from some brief quantitative answer to this question, where an advantage of PAINT is that it stays close to the real trajectory.

Minor Typos:
- (Line 19) In Abstract, replace “states parallel” with “states in parallel”.
- (Line 430) Forgot to add a closing parenthesis to the conditional probability.

**Questions:**

Most questions are embedded in the “Weaknesses” section. Some additional/ parallel ones are:

1. What is the sparse information given from the actual simulation? I.e. what precisely is m? Is it 25 randomly sampled points on the structure grid? If so, are the locations chosen randomly at initialization or is every sampling at a different random location?
2. Regarding novelty: as stated above, if my understanding is correct, novelty in this work stems from the observation that when trying to reconstruct a flow from sparse data, feeding only the sparse data (PAINT) and not the previously reconstructed time step/ time window as well (autoregressive), results in significant improvements. Could the following paper with title “Operator learning for reconstructing flow fields from sparse measurements: An energy transformer approach” Zhang et. al. be similar work to this?

---

> ### Author Response · Authors · 2025-11-19
>
> We thank the reviewer for the constructive thoughts. We will withdraw from this submission to add more experiments and further analyses. We still would like to answer your specific points from Weaknesses (W) and Questions (Q).
>
>
> W1/1: In our initial experiments we used a longer history length of the measurements, but we observed that it does not make a difference, while being computationally more expensive. We hypothesize that this is an instance of over-reliance on the autoregressive state: during training the autoregressive (teacher-forced) state $x_t$ is much more informative to predict $x_{t+1}$ than the sparse measurements. For a future submission, we plan to properly add such a line of experiments to underscore this point.
>
> W1/2: We took standard hyperparameters for the learning rate, AdamW and the LR-schedule. We did not tune the hyperparameters of any of these runs, as the observed learning curves were satisfactory.
>
> W1/3: This is an experiment we plan to add for a future submission.
>
> W1/4: There are different metrics that could be matched: parameters, FLOPS, training time, training memory. In our view comparing a model that generates “a video as a whole” (i.e. PAINT) vs. a model that generates “a video frame-by-frame” (the AR baseline) should not be FLOPS-matched. These are inherently different approaches and predicting a video as a whole requires e.g. ensuring global consistency. We argue that such an approach inherently needs more FLOPS. Therefore we argue that parameter-matching is still the best comparison.
>
> W2/1: Due to the computational constraints we trained a single model on history h=16, forward prediction n=8. We then used the last timestep for which had the measurements for reporting our results and plots. Figure 16 illustrates the full trajectory with the reconstruction of 15 past, one present and 5 of 8 future timesteps.
>
> W2/2: This is mentioned in the appendix. The resolution is 128x128. We will update this.
>
> W2/3: At training probes are uniformly sampled for each datapoint in a batch and freshly resampled for each batch. So there is no explicit reuse of the same probe locations during training. We will be more precise about this in a future revision.
>
> W3: see W2/1. Also we thank you for the suggestion and will produce such a plot for a future version.
>
> W4: Thank you, we will add a qualitative or quantitative analysis of these discontinuities.
>
> Q1: Yes, the sparse information are the 25 randomly sampled points. These points are uniformly sampled for each datapoint. The probe locations are not explicitly reused for further timesteps (there remains the small probability to sample the same 25 of 128x128=16384).
>
> Q2: PAINT differentiates the proposed work in several ways.
> First, PAINT focuses not on a specific model, but on the conceptual principles to design neural architectures for digital twins.
> Secondly, PAINT models the inherent uncertainty of the underdetermined reconstruction problem in a probabilistic way (e.g. via FlowMatching). The proposed work does not account for this uncertainty and since it’s trained with the Relative MSE, learns to reconstruct the mean of the distribution.
> Thirdly, architecturally, FlowPAINT is more general by (1) using not only spatial but also temporal information for the reconstruction and (2) FlowPAINT may use any pixel as measurements, while the mentioned model relies patches (i.e. groups of pixels).

---

### Author Response · Authors · 2025-11-19

We sincerely thank the reviewers for their time and effort.

Given the conceptual nature of our paper, we believe that extensive experiments and ablations are not essential to support the core idea presented.
However, we acknowledge the reviewers concerns regarding the comparison between our proposed framework and the baseline, as well as a lack of insightful analyses and we recognize the need to address these limitations more thoroughly.
We have therefore decided to withdraw the current submission. We will revise the work to ensure a more rigorous and fair comparison and better align the experimental evaluation with the main message of the paper.

---

### Note · Authors · 2025-11-19

I have read and agree with the venue's withdrawal policy on behalf of myself and my co-authors.